# Prognostic Value of Lymph Node-To-Primary Tumor Standardized Uptake Value Ratio in Esophageal Squamous Cell Carcinoma Treated with Definitive Chemoradiotherapy

**DOI:** 10.3390/cancers12030607

**Published:** 2020-03-06

**Authors:** Chia-Hsin Lin, Tsung-Min Hung, Yu-Chuan Chang, Chia-Hsun Hsieh, Ming-Chieh Shih, Shih-Ming Huang, Chan-Keng Yang, Ching-Fu Chang, Sheng-Chieh Chan, Wing-Keen Yap

**Affiliations:** 1Department of Radiation Oncology, Chang Gung Memorial Hospital-Linkou Medical Center and Chang Gung University, 5 Fu-Shin Street, Kwei-Shan, Taoyuan 333, Taiwan; 2Department of Nuclear Medicine and Molecular Imaging Center, Chang Gung Memorial Hospital-Linkou Medical Center and Chang Gung University, Taoyuan 333, Taiwan; 3Department of Medical Imaging and Radiological Sciences, College of Medicine, Chang Gung University, 333 Taoyuan, Taiwan; 4Circulating Tumor Cell Lab, Division of Medical Oncology, Department of Internal Medicine, Chang Gung Memorial Hospital-Linkou Medical Center and Chang Gung University, Taoyuan 333, Taiwan; 5Department of Chemical and Materials Engineering, Chang Gung University, Taoyuan 333, Taiwan; 6Institute of Epidemiology and Preventive Medicine, College of Public Health, National Taiwan University, Taipei 10617, Taiwan; 7Department of Radiation Oncology, Keelung Chang Gung Memorial Hospital, Keelung 204, Taiwan; 8Division of Hematology-Oncology, Department of Internal Medicine, Linkou Medical Center, Chang Gung Memorial Hospital, Kweishan, Taoyuan 333, Taiwan; 9Department of Nuclear Medicine, Hualien Tzu-Chi Hospital, Buddhist Tzu-Chi Medical Foundation, Hualien 970, Taiwan; 10Department of Radiation Oncology, New Taipei Municipal TuCheng Hospital (Built and Operated by Chang Gung Medical Foundation), New Taipei City 236, Taiwan

**Keywords:** FDG-PET, prognosis, ESCC, chemoradiotherapy, esophageal cancer

## Abstract

We aimed to investigate the prognostic value of the relative maximum standardized uptake value (SUV) of metastatic lymph node (LN) compared with that of primary tumor (SUV_LN_/SUV_Tumor_) based on a pretreatment [^18^F]-FDG PET/CT scan in patients with clinically node-positive esophageal squamous cell carcinoma (cN+ ESCC) treated with definitive chemoradiotherapy (dCRT). We retrospectively evaluated cN+ ESCC patients who underwent a PET/CT scan before dCRT. Time-dependent receiver operating characteristics analysis was performed to identify the optimal cutoff value for SUV_LN_/SUV_Tumor_. Prognostic influences of SUV_LN_/SUV_Tumor_ on distant metastasis-free survival (DMFS) and overall survival (OS) were evaluated using the Kaplan–Meier method and log-rank test for univariate analysis and Cox’s proportional hazards regression model for multivariate analysis. We identified 112 patients with newly diagnosed cN+ ESCC. After a median follow-up of 32.0 months, 50 (44.6%) patients had distant failure and 84 (75.0%) patients died. Patients with high SUV_LN_/SUV_Tumor_ (≥ 0.39) experienced worse outcomes than low SUV_LN_/SUV_Tumor_ (< 0.39) (two-year DMFS: 26% vs. 70%, p < 0.001; two-year OS: 21% vs. 48%, p = 0.001). Multivariate analysis showed that SUV_LN_/SUV_Tumor_ was an independent prognostic factor for both DMFS (adjusted HR 2.24, 95% CI 1.34–3.75, *p* = 0.002) and OS (adjusted HR 1.61, 95% CI 1.03–2.53, *p* = 0.037). Pretreatment of SUV_LN_/SUV_Tumor_ is a simple and useful marker for prognosticating DMFS and OS in cN+ ESCC patients treated with dCRT, which may help in tailoring treatment and designing future clinical trials.

## 1. Introduction

Patients with esophageal cancer often have dismal prognosis, with poor 5-year overall survival rates estimated to be 15–25% [1,2]. In Asia, esophageal squamous-cell carcinoma (ESCC) remains the predominant histology, and it has been considered to be more responsive to chemoradiotherapy (CRT) than adenocarcinoma histology, with higher pathological response rate in ESCC following neoadjuvant CRT [3,4,5]. For locally advanced ESCC, there have been two phase III prospective trials suggesting that definitive chemoradiotherapy (dCRT) resulted in equivalent overall survival (OS) to trimodality therapy (neoadjuvant CRT followed by a planned surgery) [6,7,8]. Trimodality therapy achieved improved local tumor control at the price of higher treatment-related death. Furthermore, distant metastasis (DM) rate remained similarly high between two treatment approaches and was reported as 30–40% at two years, which substantially restricted patients’ survival. To eradicate micrometastasis and improve survival outcomes, the role of induction chemotherapy has been studied in recent trials, while showing conflicting results with respect to the survival benefit from induction chemotherapy [9,10,11,12]. Consequently, it would be of great clinical value to identify risk factors for DM, which might help select at-risk patients who may gain advantages from more aggressive treatments in future clinical trials.

Positron-emission-tomography (PET) with [^18^F]-fluorodeoxyglucose (FDG) has recently been more widely applied in the management of locally advanced esophageal cancer, particularly in identifying occult DM in pretreatment staging and detection of interval metastasis [13]. Furthermore, recent studies have demonstrated the prognostic values of standardized uptake value (SUV) of the metastatic lymph nodes (LNs), in addition to the SUV of primary tumor [14,15,16,17,18,19], on both pretreatment [20] and posttreatment PET scans [21] in patients treated with dCRT. Intriguingly, the LN-to-tumor SUV ratio (SUV_LN_/SUV_Tumor_) recently has been shown to have a strong relationship with clinical outcomes in other disease sites, including lung [22], pancreatic [23], breast [24], and gynecologic cancers [25,26]. However, there is currently no published study investigating the relative SUVs of metastatic LNs to that of primary tumor in ESCC. 

The purpose of this study was to determine if the SUV_LN_/SUV_Tumor_ evaluated by pretreatment FDG-PET has prognostic values, particularly in identifying patients at risk of developing distant metastasis after treatment, in patients with ESCC undergoing dCRT.

## 2. Results

### 2.1. Patient and Treatment Characteristics

A total of 112 patients with newly diagnosed ESCC at our institution were identified and enrolled in the study according to the predetermined inclusion and exclusion criteria (Figure 1). Patient and treatment characteristics are summarized in Table 1 and Figure 1. Of the 112 patients in the cohort, 97.3% were male, and the median age was 56 years (IQR, 50–62 years). Over 63% of patients had AJCC stage IIIC disease, and median tumor length was 6.0 cm (IQR, 4.9–8.0). The most common tumor location was upper third (46.4%) and middle third (44.6%). The median values (IQR) of SUV_Tumor_, SUV_LN_, and SUV_LN_/SUV_Tumor_ were 17.7 (14.2–23.9), 9.9 (4.2–15.3), and 0.59 (0.25–0.78), respectively. Of 79 patients who received a radiotherapy dose < 5000 cGy, 59.5% (*n* = 47) patients underwent consolidative CRT, while 40.5% (*n* = 32) patients did not undergo consolidative CRT due to interval metastasis, medically unfit, or patient refusal.

### 2.2. Measurement of Cutoff Values for FDG-PET Parameters

In terms of SUV_LN_/SUV_Tumor_, time-dependent ROC analysis identified an optimal cutoff value of 0.39 for DMFS (area under the curve 0.754; *p* < 0.01; 95% CI 0.593–0.916; Figure 2), and the sensitivity and specificity at this value was 74.2% and 80.0%, respectively. To evaluate the potential influence on the optimal cutoff value of SUV_LN_/SUV_Tumor_ from using two different PET/CT scanners, we conducted a sensitivity analysis by performing time-dependent ROC curve analysis separately on patients using different scanners. For patients using the Discovery ST16 scanner (*n* = 62) and Biograph mCT scanner (*n* = 50), the optimal cutoff values of SUV_LN_/SUV_Tumor_ were 0.38 and 0.39, respectively, and the IQRs of SUV_LN_/SUV_Tumor_ were 0.22–0.74 and 0.28–0.83, respectively. The variation of the optimal cutoff values by using the two different PET/CT scanners was negligible, particularly in the context of the wide spread of the SUV_LN_/SUV_Tumor_ values. As a comparison, the optimal cutoff values of SUV_LN_ for patients using the Discovery ST16 scanner and Biograph mCT scanner were 6.31 and 7.55, respectively, and the IQRs were 3.69–15.30 and 4.51–15.16, respectively, suggesting that the non-normalized SUV_LN_ parameter suffered much greater inter-scanner variability.

Thus, according to the ROC curve analysis, the patients were separated into low SUV_LN_/SUV_Tumor_ (< 0.39) and high SUV_LN_/SUV_Tumor_ (≥ 0.39).

### 2.3. Survival Analyses

After a median follow-up of 32.0 months (95% CI: 26.9–37.1 months), out of 112 patients treated with dCRT, 50 (44.6%) patients had distant failure and 84 (75.0%) patients died. Kaplan–Meier estimates showed that patients with high SUV_LN_/SUV_Tumor_ had worse outcomes than low SUV_LN_/SUV_Tumor_ (two-year DMFS: 26% vs. 70%, *p* < 0.001; two-year OS: 21% vs. 48%, *p* = 0.001) (Figure 3).

### 2.4. Correlations between Parameters Evaluated by FDG-PET and Clinicopathological Features

Table 2 shows the association between PET-derived parameters and clinicopathological features. While advanced clinical nodal stage (r = 0.362, *p* < 0.001) and SUV_Tumor_ (r = −0.223, *p* = 0.018) showed weak correlations with SUV_LN_/SUV_Tumor_, SUV_LN_ (r = 0.744, *p* < 0.001) appeared to be strongly correlated with SUV_LN_/SUV_Tumor_ and thus should be excluded from multivariate Cox regression analysis to avoid multicollinearity effect [27].

### 2.5. Evaluation of Prognostic Factors of DMFS and OS

Table 3 summarizes the results of the Cox proportional hazard model of prognostic factors for DMFS and OS in the present study. The univariate model contained eight clinical and three PET-derived parameters (age, performance status, tumor location, initial T stage, initial N stage, tumor length, chemotherapy regimens, radiotherapy dose, SUV_Tumor_, SUV_LN_, and SUV_LN_/SUV_Tumor_). Univariate Cox regression revealed initial N stage (*p* = 0.001), SUV_Tumor_ (*p* = 0.001), SUV_LN_ (*p* < 0.001), and SUV_LN_/SUV_Tumor_ (*p* = 0.002) as significant prognostic factors for DMFS. SUV_LN_ (*p* = 0.003), as well as SUV_LN_/SUV_Tumor_ (*p* = 0.026) were significant prognostic factors for OS. Since there was multicollinearity between SUV_LN_ and SUV_LN_/SUV_Tumor_ (*r* = 0.744), SUV_LN_ was not included in the multivariate Cox model [27]. After being adjusted with potential confounders with multivariate Cox regression analysis, SUV_LN_/SUV_Tumor_ remained as a strong independent adverse prognostic factor for both DMFS (adjusted HR 2.24, 95% CI 1.34–3.75, *p* = 0.002) and OS (adjusted HR 1.61, 95% CI 1.03–2.53, *p* = 0.037). Furthermore, age (adjusted HR 0.96, 95% CI 0.92–0.99, *p* = 0.020), nodal stage N3 (adjusted HR 2.21, 95% CI 1.18–4.14, *p* = 0.013), and SUV_Tumor_ (adjusted HR 1.08, 95% CI 1.04–1.12, *p* < 0.001) were independent prognostic factors for DMFS; while tumor stage T4 (adjusted HR 1.66, 95% CI 1.06–2.60, *p* = 0.028) was an independent risk factor for OS.

## 3. Discussion

Esophageal cancer with squamous cell carcinoma histology is more sensitive to chemoradiation than adenocarcinoma [3,4,5]. Two European phase III randomized controlled trials have provided evidence that dCRT resulted in comparable survival times to preoperative CRT plus surgery [6,7,8]. Therefore, dCRT has been regarded as a valid option of definitive treatment for patients with locally advanced ESCC in ESMO (European Society for Medical Oncology) guideline [28]. However, despite multimodality treatment with dCRT, substantial patients still succumb to distant recurrences [6,7,8,20], which thereby warrants the need to explore more aggressive treatments. In theory, the addition of induction chemotherapy potentially imparts beneficial effects due to early elimination of micrometastasis, enhancement of sensitivity to the subsequent CRT, and allowance for enough time for careful radiotherapy planning [29,30,31]. Several prospective, single arm trials have investigated the feasibility of induction chemotherapy preceding dCRT for ESCC. In INT 0122 trial, Minsky et al. reported that induction chemotherapy with cisplatin and fluorouracil followed by concurrent CRT for clinical stage T1-4N0-1M0 ESCC (*n* = 38) yielded a complete response rate of 47% and median OS of 20 months [32]. In phase II FFCD trial, induction cisplatin-irinotecan before CRT without surgery for stage I–III esophageal cancer (*n* = 43) resulted in a complete clinical response rate of 58.1% and 1-year OS rate of 62.8% [33]. Additionally, Satake et al. showed remarkable results in a Japanese multicenter phase I/II study for unresectable ESCC patients with a 3-year OS rate of up to 40.4% [34]. Nevertheless, the value of administration of induction CT before dCRT is still controversial due to the lack of prospective phase III randomized trials. Furthermore, it has been suggested that induction chemotherapy might only be of benefit in high-risk ESCC patients [29] due to the fact that no clear survival advantage has been shown in prospective studies in which most patients were esophageal adenocarcinoma and presented with earlier stage diseases [12,35]. These facts thus underscore the importance of the purpose of this study, which is using the SUV_LN_/SUV_Tumor_ of pretreatment PET to identify patients at high-risk for distant failures after receiving dCRT, who might derive the greatest benefits from adding induction chemotherapy before dCRT.

The clinical implications of SUV_LN_/SUV_Tumor_ have been investigated in recent studies in a wide variety of cancers [22,23,24,25,26,36,37,38]. The first established value for its use is to help evaluate the presence of metastatic LN. Cerfolio et al. reported that the SUV_LN_/SUV_Tumor_ of 0.56 could predict mediastinal nodal pathology in patients with nonsmall cell lung cancer, with significantly higher area under curve than SUV_Tumor_ [37]. Park et al. also showed that SUV_LN_/SUV_Tumor_ better predicted the presence of axillary LN metastasis than SUV_LN_ in breast cancer [38]. Based on these results, several studies further examined the role of SUV_LN_/SUV_Tumor_ in prognostication of clinical outcomes. It has been demonstrated that higher SUV_LN_/SUV_Tumor_ portended lower response to initial chemotherapy and poorer survival (PFS and OS) for patients with nonsmall cell lung cancer [22]. Similarly, it has been observed that SUV_LN_/SUV_Tumor_ was an independent covariate for predicting relapse in cervical squamous cell carcinoma [26], invasive ductal breast cancer [24], and resectable pancreatic cancer [23]. 

To date, the prognostic value of SUV_LN_/SUV_Tumor_ in ESCC has not been reported. The purpose of this study was to investigate the prognostic significance of relative metabolic activity of metastatic LNs compared with that of primary tumor in patients with cN+ ESCC treated with dCRT. To the best of our knowledge, this is the first study to report the prognostic value of SUV_LN_/SUV_Tumor_ in ESCC. This study revealed that ESCC patients with higher pretreatment of SUV_LN_/SUV_Tumor_ had significantly shorter distant metastasis-free survival time after dCRT than ESCC patients with lower pretreatment of SUV_LN_/SUV_Tumor_. Moreover, SUV_LN_/SUV_Tumor_ was an independent factor for predicting both DMFS and OS after being adjusted with potential confounding factors. These results underscore the significance of relative metabolic activity of metastatic LNs versus that of primary tumor, suggesting SUV_LN_/SUV_Tumor_ could be a promising prognostic indicator for ESCC patients before receiving dCRT.

We previously reported the prognostic value of pretreatment SUVmax of the metastatic lymph nodes in patients with ESCC treated with dCRT [20,21]. However, there are some well-known drawbacks of using nonnormalized SUV, e.g., partial volume effect, uptake time dependence of the SUV, and interstudy variability of image acquisition and reconstruction parameters [26,39,40,41,42], which possibly undermine the reliability of SUV, thereby limiting its prognostic usefulness for esophageal cancer patients. The strength of the current study is that the SUV_LN_/SUV_Tumor_ is simple to calculate and may have less inter-scanner variability compared to the SUV_LN_, and thus SUV_LN_/SUV_Tumor_ may have better generalizability. In addition, SUV_LN_/SUV_Tumor_ may reflect the aggressiveness of the primary tumor and the biological interactions between primary tumor and metastatic LNs as indicated by previous studies in other disease sites [22,23,24,26].

The major limitation of our study is its retrospective nature and that it was conducted at a single institution with suboptimal sample size. This may introduce inherent selection bias and therefore limit the generalizability of our findings. Certainly, further validation of the prognostic value of SUV_LN_/SUV_Tumor_ by prospective and large studies is needed before its common use in clinical practice. Nevertheless, the current study is still noteworthy as it is the first study to report the prognostic impact of SUV_LN_/SUV_Tumor_ in patients with ESCC treated by dCRT. This study provides insights to the importance of further studying the pretreatment relative metabolic activity of metastatic lymph nodes and serves as a reference for future trial designs.

## 4. Materials and Methods 

### 4.1. Patient Selection

This study involved patients who were newly diagnosed with ESCC and underwent dCRT between 2009 and 2017 from the prospectively acquired database in the cancer registry of our hospital. The inclusion criteria were the following: Histopathologically confirmed ESCC, with pretreatment FDG-PET/CT, clinically detected metastatic LNs (cN+), and radiation dose at least 4000 cGy delivered with continuous fashion, i.e., not split-course, with concurrent chemotherapy as the primary treatment. The exclusion criteria included the presence of DM at diagnosis and the presence of another cancer diagnosis prior to CRT. The pretreatment staging examinations included esophagogastroduodenoscopy (EGD) with biopsies, endoscopic ultrasound (EUS), chest and abdominal contrast-enhanced CT (CECT), and FDG-PET/CT. Clinical staging was based on the American Joint Committee on Cancer (AJCC), 7th edition [43]. The method of detection of lymph nodes in this study was based on both CECT and FDG-PET/CT. Any discrepancies between the CECT and FDG-PET/CT results were resolved through the consensuses in multidisciplinary team meetings.

The institutional review board of our hospital has approved this study (no. 201900883B0). All participants included in the study signed informed consent forms for treatment before the start of study treatments.

### 4.2. Treatment

All patients were treated with dCRT according to the treatment guideline of our institution. This consisted of radiotherapy in the form of external radiotherapy (prescribed dose at least 4000 cGy delivered in a continuous fashion) with concurrent chemotherapy. The chemotherapy regimens encompassed cisplatin and 5-fluorouracil, paclitaxel and carboplatin, or paclitaxel and cisplatin. After completing dCRT, all patients underwent tumor response assessment using chest and abdominal CECT, EGD/EUS with or without biopsy, and FDG-PET if possible. For patients who received radiotherapy dose < 5000 cGy, consolidative CRT of 2000–2300 cGy was further advised if patients were eligible (medically fit, no evidence of interval metastasis on restaging imaging, and the patients consent to additional CRT), adapting from the protocol of the FFCD 9102 phase III study [8]. The variability of radiotherapy dose was mainly due to the evolution of the treatment guidelines over time.

### 4.3. FDG-PET/CT

Pretreatment PET/CT scans were arranged for the staging purpose. Each patient was asked to fast for at least 4 h prior to examination. One hour before imaging, patients were administered intravenously with 200–444 MBq of [^18^F] FDG, depending on body weight. PET/CT was performed on a Discovery ST16 scanner (GE Healthcare) or a Biograph mCT scanner (Siemens Medical Solution). Nonenhanced CT was performed to generate an attenuation correction map for PET, and the PET images were reconstructed by an ordered-subset expectation maximization iterative reconstruction algorithm (four iterations and 10 subsets for the Discovery ST16; two iterations and 21 subsets for the Biography mCT). Both PET and CT scan were acquired from the skull base to the midthigh. FDG-PET images were evaluated by an experienced nuclear medicine physician. The maximum standardized uptake value (SUVmax) was quantitatively applied to represent [^18^F] FDG avidity. SUV was calculated as adjusted with body weight and radioactive decay at scanning time using the formula as follows: SUV = activity concentration/[injected dose/bodyweight]). The SUVmax of the primary tumor and metastatic LN were separately measured, which subsequently generated PET parameters such as SUVmax of the primary ESCC (SUV_Tumor_), SUV_LN_, and the LN-to-primary ESCC SUV ratio (SUV_LN_/SUV_Tumor_). The SUV_LN_/SUV_Tumor_ was generated by dividing the SUVmax of the lymph node with the highest FDG consumption by the SUVmax of the primary tumor.

### 4.4. Post-Therapy Surveillance and Clinical Endpoints

Patients following dCRT were routinely followed up every three months during the first and second years, every 4–6 months during the third and fourth years, and every 6–12 months thereafter. Radiographic surveillance was arranged according to the protocol of our institution: CT scans were performed every 3–6 months and PET/CT was performed during the first time of follow-up or when the patients had symptoms suggesting recurrence. EGD was arranged every 3–6 months or when recurrence was suspected. The primary endpoint was distant metastasis-free survival (DMFS) and the secondary endpoint was OS. The endpoints in this study were calculated from the date of pathological diagnosis.

### 4.5. Statistical Analysis

Baseline characteristics of study subjects were summarized as frequencies and percentages for categorical data, and medians and interquartile ranges (IQR) for continuous data. The median follow-up time was estimated by the reverse Kaplan–Meier method [44]. Time-dependent receiver operating characteristic (ROC) curve analysis according to the median follow-up time [45] was employed to determine the optimal cutoff value of SUV_LN_/SUV_Tumor_ for discriminating the patients based on DMFS. The optimal cutoff value was determined by the point on the time-dependent ROC curve closest to (0,1). The DMFS and OS curves were estimated by the Kaplan–Meier method, and the p-values were determined by the log-rank test. Multicollinearity between variables was estimated using the Pearson correlation coefficient [27]. The independent influences of various prognostic factors were analyzed by Cox’s proportional hazards regression model and presented as hazard ratios (HRs) and 95% confidence intervals (CIs). All prognostic factors with *p*-values < 0.1 in the univariate model were further entered into the multivariate analysis. All the statistical tests were two-sided and a *p*-value of < 0.05 was defined as statistically significant. Analyses were performed with the use of IBM SPSS statistical software (version 21; SPSS, Inc., Chicago, IL). The time-dependent ROC was created using the R software “ROCt” package (version 3.6.0).

## 5. Conclusions

This study demonstrated for the first time that SUV_LN_/SUV_Tumor_ on pretreatment [^18^F]-FDG-PET was an independent prognosticator for DMFS and OS for patients with cN+ ESCC receiving dCRT. Our results suggest that relative metabolic activity of metastatic LN on FDG-PET may become a useful indicator for distant recurrence and survival before dCRT. This novel and promising parameter may help the design of prospective clinical trials, and the individual tailoring of treatment. 

## Figures and Tables

**Figure 1 cancers-12-00607-f001:**
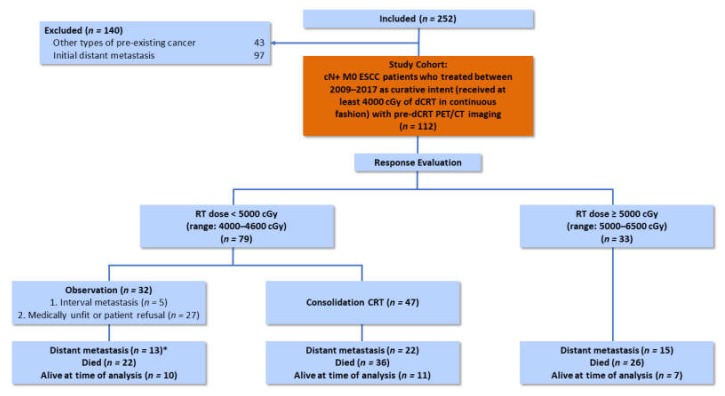
Flowchart of treatment characteristics of eligible patients through the study. cN+ M0 ESCC: Clinically node-positive nonmetastatic esophageal squamous cell carcinoma; CRT: Chemoradiotherapy; dCRT: Definitive chemoradiotherapy; RT: Radiotherapy. *This included the five patients who did not undergo consolidative CRT due to the detection of interval metastasis.

**Figure 2 cancers-12-00607-f002:**
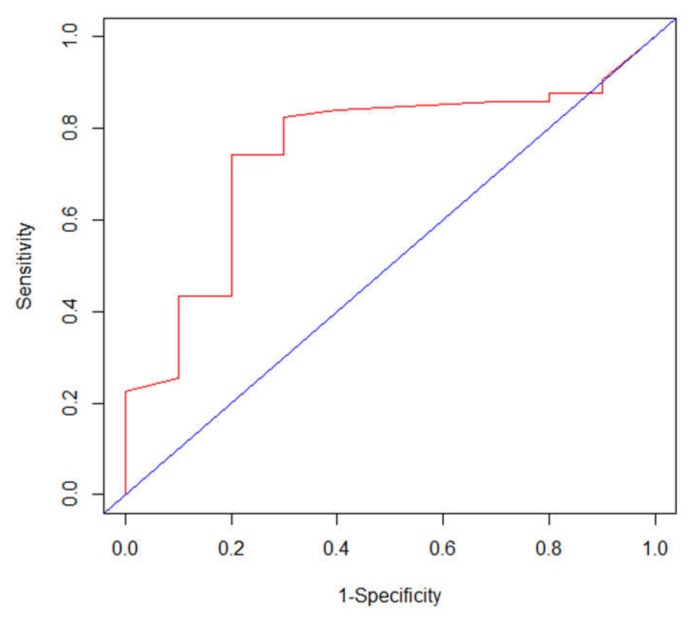
Time-dependent receiver operating characteristic curve analysis of distant metastasis-free survival prediction based on the SUV_LN_/SUV_Tumor_ ratio in 112 patients with esophageal squamous cell carcinoma (ESCC). The area under the curve was 0.754 (*p* < 0.01, 95% CI 0.593–0.916), and 0.39 was determined as the best SUV_LN_/SUV_Tumor_ ratio cutoff value for survival prediction.

**Figure 3 cancers-12-00607-f003:**
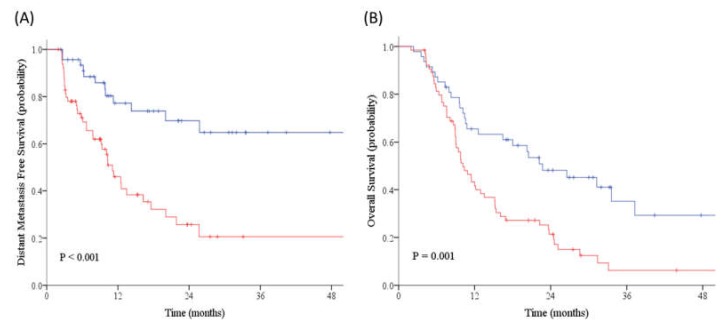
The Kaplan–Meier estimates of (**A**) distant metastasis-free survival (DMFS) and (**B**) overall survival (OS) for patients with SUV_LN_/SUV_Tumor_ ≥ 0.39 (red line) versus SUV_LN_/SUV_Tumor_ < 0.39 (blue line). High SUV_LN_/SUV_Tumor_ (≥ 0.39) predicted for worse outcomes than low SUV_LN_/SUV_Tumor_ (< 0.39) on DMFS (two-year: 26% vs. 70%, *p* < 0.001) and OS (two-year: 21% vs. 48%, *p* = 0.001).

**Table 1 cancers-12-00607-t001:** Baseline characteristics of patients who underwent positron emission tomography/computed tomography before dCRT for esophageal squamous cell carcinoma.

Characteristics	Patients	%
Median age, years (IQR)	56 (50–62)	
Male sex	109	97.3
ECOG		
0	20	17.9
1	89	79.5
2	3	2.7
Tumor location		
Upper	52	46.4
Middle	50	44.6
Lower	10	8.9
cT classification ^a^		
T1	2	1.8
T2	10	8.9
T3	48	42.9
T4	52	46.4
cN classification ^a^		
N1	19	17.0
N2	55	49.1
N3	38	33.9
cStage^a^		
IIB	3	2.7
IIIA	13	11.6
IIIB	25	22.3
IIIC	71	63.4
Median tumor length, cm (IQR)	6.0 (4.9–8.0)	
Median SUV_Tumor_ (IQR)	17.7 (14.2–23.9)	
Median SUV_LN_ (IQR)	9.9 (4.2–15.3)	
Median SUV_LN_/SUV_Tumor_ (IQR)	0.59 (0.25–0.78)	
Chemotherapy		
Carboplatin/Paclitaxel	63	56.2
Cisplatin/5-FU	46	41.1
Cisplatin/Paclitaxel	3	2.7
Median total RT dose, cGy (IQR)	6000 (4500–6480)	

dCRT: Definitive chemoradiotherapy; ECOG: Eastern Cooperative Oncology Group performance score; 5-FU: 5-Fluorouracil; IQR: Interquartile range; LN: Lymph node; RT: Radiotherapy; SUV: Standardized uptake value. ^a^ Clinical staging according to Tumor-Node-Metastasis classification, 7th edition of the American Joint Committee on Cancer staging system.

**Table 2 cancers-12-00607-t002:** Correlations between SUV_LN_/SUV_Tumor_ and other clinical prognostic factors.

Characteristics	SUV_LN_/SUV_Tumor_	
	Correlation Coefficient ^a^	*P*-Value
Age	−0.023	0.810
Tumor location	−0.090	0.344
cT classification ^b^	−0.123	0.198
cN classification ^b^	0.362	< 0.001
Tumor length	0.004	0.968
Chemotherapy	0.032	0.739
SUV_Tumor_	−0.223	0.018
SUV_LN_	0.744	< 0.001

LN: Lymph node; SUV: Standardized uptake value. ^a^ Pearson correlation coefficient method was applied. ^b^ Clinical staging according to TNM classification, 7th edition.

**Table 3 cancers-12-00607-t003:** Univariate and multivariate analysis of risk factors associated with DMFS and OS in ESCC patients treated with dCRT.

	DMFS				OS			
Predictive Variables	Univariate Analysis		Multivariate Analysis ^a^		Univariate Analysis		Multivariate Analysis ^a^	
	HR (95% CI)	*P*-Value	HR (95% CI)	*P*-Value	HR (95% CI)	*P*-Value	HR (95% CI)	*P*-Value
Age, years	0.97 (0.94–1.01)	0.092	0.96 (0.92–0.99)	0.020	1.01 (0.98–1.03)	0.589		
ECOG								
0 vs. 1/2 (ref)	1.02 (0.52–2.00)	0.948			0.83 (0.49–1.42)	0.499		
Tumor location								
Upper vs.	0.93 (0.53–1.63)	0.806			0.83 (0.54–1.27)	0.393		
Middle/Lower (ref)								
Initial T-Stage ^b^								
cT4 vs. cT1–3 (ref)	1.11 (0.63–1.94)	0.720			1.45 (0.94–2.22)	0.094	1.66 (1.06–2.60)	0.028
Initial N-Stage ^b^								
cN3 vs. cN1–2 (ref)	2.60 (1.48–4.56)	0.001	2.21 (1.18–4.14)	0.013	1.51 (0.97–2.36)	0.067	1.32 (0.82–2.14)	0.252
Tumor length, cm	1.04 (0.94–1.16)	0.450			1.04 (0.95–1.13)	0.424		
SUV_Tumor_	1.06 (1.02–1.09)	0.001	1.08 (1.04–1.12)	< 0.001	1.01 (0.98–1.04)	0.488		
SUV_LN_	1.10 (1.06–1.14)	< 0.001			1.04 (1.02–1.07)	0.003		
SUV_LN_/SUV_Tumor_	1.99 (1.29–3.05)	0.002	2.24 (1.34–3.75)	0.002	1.57 (1.06–2.35)	0.026	1.61 (1.03–2.53)	0.037
Chemotherapy								
Paclitaxel/Cisplatin or Carboplatin	1.22 (0.69–2.15)	0.495			0.96 (0.62–1.49)	0.848		
Cisplatin/5-FU (ref)								
Radiotherapy		0.533				0.321		
Initial dose < 5000 cGy without consolidative boost	1.47 (0.70–3.09)	0.315			1.54 (0.86–2.76)	0.145		
Initial dose < 5000 cGy with consolidative boost	1.04 (0.54–2.00)	0.918			1.13 (0.67–1.91)	0.640		
Initial dose ≥ 5000 cGy (ref)								

CI: Confidence interval; dCRT: Definitive chemoradiotherapy; DMFS: Distant metastasis-free survival; ECOG: Eastern Cooperative Oncology Group performance score; 5-FU: 5-Fluorouracil; HR: Hazard ratio; LN: Lymph node; OS: Overall survival; SUV: Standardized uptake value. ^a^ Due to multicollinearity (*r* > 0.70) between SUV_LN_/SUV_Tumor_ and SUV_LN_, SUV_LN_ was not included in the multivariate Cox model. All other factors with *p* < 0.1 in the univariate analysis were included in the Cox multivariate analysis. ^b^ Clinical staging according to TNM classification, 7th edition.

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
