# Peer review of "Prognostic Value of Lymph Node-To-Primary Tumor Standardized Uptake Value Ratio in Esophageal Squamous Cell Carcinoma Treated with Definitive Chemoradiotherapy"

_cancers, 2020, doi:10.3390/cancers12030607_

Round 1

Reviewer 1 Report

The significance of this data is only if it can be a prospective marker and alter treatment [planning

How does it perform compared to early PET after say one cycle of chemotherapy in predicting outcome. Clearly one PET would be better than two to stratify for treatment

Author Response

Comments to the Author:

Comment #1

The significance of this data is only if it can be a prospective marker and alter treatment planning

Reply #1

We agree with the reviewer’s comment that the significance of the data depends on whether SUVLN / SUVTumor can be a pretreatment marker that may alter our treatment planning in clinical practice. However, due to its retrospective nature, the study results can only be hypothesis-generating and further validation of the prognostic significance of SUVLN / SUVTumor by prospective and large studies is definitely needed. Nevertheless, the current study is still noteworthy as it is the first study to report the prognostic impact of SUVLN / SUVTumor in patients with ESCC treated by dCRT. This study provides insights to the importance of further studying of the pretreatment relative metabolic activity of metastatic lymph nodes and serves as a reference for future trial designs.

We stated the above mentioned limitation of our study in section 3. Discussion, page 10, lines 229-234: “Certainly, further validation of the prognostic value of SUVLN / SUVTumor by prospective and large studies is needed before its common use in clinical practice. Nevertheless, the current study is still noteworthy as it is the first study to report the prognostic impact of SUVLN / SUVTumor in patients with ESCC treated by dCRT. This study provides insights to the importance of further studying of the pretreatment relative metabolic activity of metastatic lymph nodes and serves as a reference for future trial designs.”

Comment #2
How does it perform compared to early PET after say one cycle of chemotherapy in predicting outcome. Clearly one PET would be better than two to stratify for treatment.   

Reply #2

We sincerely thank you for the valuable question. However, the purpose of this study was to identify patients at high-risk for distant failures after receiving definitive chemoradiotherapy (dCRT) who might derive the greatest benefits from adding induction chemotherapy before dCRT. Thus, this study only included patients treated with upfront dCRT. Therefore, we did not have the relevant data regarding early PET after one cycle of chemotherapy in the setting of induction chemotherapy. The comparison of the prognostic values of SUVLN / SUVTumor between the two PETs (pretreatment PET and early PET) might need future studies to explore.

Reviewer 2 Report

The authors explored the significance of lymph node-to-primary tumour SUV value ratio in ESCC patients treated with definitive chemoradiotherapy. Evaluating the SUV ratio is easy and it would be useful if its clinical utility is proven. However, there are some points to be reconsidered in this study. 

#1. Authors should define 'definitive chemoradiotherapy'. This study included patients who received less than 50 gray of radiation, but radiation dose less than 50.0Gy is generally not considered as 'definitive' therapy. Also, a total administered dose of concurrent chemotherapy might be lower with these patients who received lower radiation dose. These factors might be associated with shorter prognosis. 

#2. Authors may describe any clinical suggestion of treatment for patients with high risk feature.  

Author Response

Comments to the Author:

The authors explored the significance of lymph node-to-primary tumour SUV value ratio in ESCC patients treated with definitive chemoradiotherapy. Evaluating the SUV ratio is easy and it would be useful if its clinical utility is proven. However, there are some points to be reconsidered in this study.

Comment #1

Authors should define 'definitive chemoradiotherapy'. This study included patients who received less than 50 gray of radiation, but radiation dose less than 50.0Gy is generally not considered as 'definitive' therapy. Also, a total administered dose of concurrent chemotherapy might be lower with these patients who received lower radiation dose. These factors might be associated with shorter prognosis.

Reply #1

Thank you for bringing up this important issue. We defined the definitive chemoradiotherapy cohort according to the treatment guideline of our institution in subsection 4.2. Treatment, page 10, lines 254-264. The variability of radiotherapy dose was mainly due to the evolution of the treatment guidelines over time. Before the change of treatment protocol to administration of radiotherapy dose of >50 Gy in continuous fashion with concurrent chemotherapy (the new protocol), the old treatment protocol for definitive chemoradiotherapy in our intuition was, adapted from FFCD 9102 trial protocol, administration of radiotherapy dose of >40 Gy in continuous fashion with concurrent chemotherapy followed by a treatment break for treatment response evaluation before an additional consolidative chemoradiation of 20–23 Gy. Indeed, a portion of the study cohort who were treated with the old treatment protocol did not received the additional consolidative chemoradiation for reasons stated in the manuscript (patient’s refusal, medically unfit and/or evidence of interval metastasis on treatment response evaluation). However, these patients were prospectively enrolled into the cancer registry of our institution as receiving definitive chemoradiotherapy in the spirit of intent-to-treat.

Also, we included the types of radiotherapy received as a variable in the cox regression analysis (Table 3). There was no statistically significant difference in terms of DMFS and OS for patients who received < 5000cGy of radiotherapy with no consolidative CRT comparing to that of patients who were treated with the new treatment protocol (Initial dose ≥ 5000 cGy). Nevertheless, data from phase 3 RCTs in patients treated with neoadjuvant CRT followed by surgery showed that CRT of 4000 –4140 cGy (reference: 1. NEOCRTEC5010 trial (Yang, J Clin Oncol 2018); 2. CROSS trial (Shapiro, Lancet Oncol 2015)) and CRT of 5040 cGy (reference: CALGB 9781 trial (Tepper, J Clin Oncol 2008)) resulted in similar pathologic complete rate of 40%–50% for squamous cell carcinoma of the esophagus. This add to the argument that 4000 cGy of CRT may not be inferior to 5000 cGy of CRT. Furthermore, as far as the authors acknowledged, to date there is no randomized controlled trial’s data that 4000 cGy of CRT is inferior to 5000 cGy of CRT in terms of tumor control and survival in the setting of definitive chemoradiotherapy for patients with esophageal squamous cell carcinomas.

Comment #2

Authors may describe any clinical suggestion of treatment for patients with high risk feature.

Reply #2

Thank you for your valuable suggestion. We added the following sentence in the manuscript to address this issue: (refer to section 3. Discussion, page 9, lines 190-193,) “These facts thus underscore the importance of the purpose of this study, which is using the SUVLN / SUVTumor of pretreatment PET to identify patients at high-risk for distant failures after receiving dCRT, who might derive the greatest benefits from adding induction chemotherapy before dCRT.

Reviewer 3 Report

This study used retrospective data of pretreatment [18F]-FDG PET/CT scan in esophageal squamous cell carcinoma (ESCC) patients and revealed a reverse association between the ratio of maximum standardized uptake value (SUV) of metastatic lymph node (LN) over that of primary tumor (SUVLN/SUVTumor) and distant metastasis-free survival (DMFS) or overall survival (OS). The authors suggest that the pretreatment SUVLN/ SUVTumor is a better marker for prognosticating DMFS and OS in ESCC patients. The statistical analysis is appropriate, and the data support the conclusion to certain extend. There are some critiques.    

  1. Published study has shown that SUVLN has significant positive correlations with pathological stage, LN status and differentiation (Dong et al., Mol Imaging Biol. 2019 Feb;21(1):175-182) in ESCC patients. Therefore, this study has less novelty. The authors argued that the ratio of SUVLN/SUVtumor is better than SUVLN since it can overcome the drawbacks of using nonnormalized SUV, e.g., partial volume effect, uptake time dependence of the SUV, and interstudy variability of image acquisition and reconstruction parameters. Yet, there is no data to support this point. A detailed comparative study of the reliability of SUVLN/SUVtumor and SUVLN should be conducted.
  2. The inflamed tissue including tumor-draining LNs can also show high uptake of [18F]FDG. Did the uptake measurements for LNs verified by postoperative pathological examination to reduce the risk of false-positive results? Such data should be presented.
  3. Table 2, why “Age” is shown in Bold font with underline?

Author Response

Comments to the Author:

This study used retrospective data of pretreatment [18F]-FDG PET/CT scan in esophageal squamous cell carcinoma (ESCC) patients and revealed a reverse association between the ratio of maximum standardized uptake value (SUV) of metastatic lymph node (LN) over that of primary tumor (SUVLN/SUVTumor) and distant metastasis-free survival (DMFS) or overall survival (OS). The authors suggest that the pretreatment SUVLN/ SUVTumor is a better marker for prognosticating DMFS and OS in ESCC patients. The statistical analysis is appropriate, and the data support the conclusion to certain extend. There are some critiques.

Comment #1

Published study has shown that SUVLN has significant positive correlations with pathological stage, LN status and differentiation (Dong et al., Mol Imaging Biol. 2019 Feb;21(1):175-182) in ESCC patients. Therefore, this study has less novelty. The authors argued that the ratio of SUVLN/SUVtumor is better than SUVLN since it can overcome the drawbacks of using nonnormalized SUV, e.g., partial volume effect, uptake time dependence of the SUV, and interstudy variability of image acquisition and reconstruction parameters. Yet, there is no data to support this point. A detailed comparative study of the reliability of SUVLN/SUVtumor and SUVLN should be conducted.

Reply #1

Thank you for pointing out this issue. To show the potential advantages of using SUVLN / SUVTumor over SUVLN, we further performed sensitivity analysis to examine inter-scanner variability of these two parameters. We conducted time-dependent ROC curve analysis separately on patients using different scanners to investigate the potential influence on the optimal cut-off value of SUVLN / SUVTumor and SUVLN from using two different PET/CT scanners. For patients using Discovery ST16 scanner (n = 62) and Biograph mCT scanner (n = 50), the optimal cut-off values of SUVLN / SUVTumor were 0.38 and 0.39 respectively, and the IQRs of SUVLN / SUVTumor were 0.22–0.74 and 0.28–0.83 respectively. The variation of the optimal cut-off values by using the two different PET/CT scanners was negligible, particularly in the context of the wide spreads of the SUVLN / SUVTumor values. As a comparison, the optimal cut-off values of SUVLN for patients using Discovery ST16 scanner and Biograph mCT scanner were 6.31 and 7.55 respectively, and the IQRs were 3.69–15.30 and 4.51–15.16 respectively, suggesting that the non-normalized SUVLN parameter suffered much greater inter-scanner variability. We added the following sentences in section 2.2. Measurement of cutoff values for FDG-PET parameters, page 4, lines 114-118: “As a comparison, the optimal cut-off values of SUVLN for patients using Discovery ST16 scanner and Biograph mCT scanner were 6.31 and 7.55 respectively, and the IQRs were 3.69–15.30 and 4.51–15.16 respectively, suggesting that the non-normalized SUVLN parameter suffered much greater inter-scanner variability.”

Nevertheless, We agree with the reviewer that there is indeed no solid data to support that SUVLN / SUVTumor can overcome the drawback of partial volume effect when using nonnormalized SUVLN. Thus, we removed the phrase “may address the drawback of partial volume effect when using a non-normalized SUV parameter such as the SUVmax of the metastatic lymph nodes” and replaced it with the phrase “may have less inter-scanner variability compared to the SUVLN” in the section 3. Discussion, page 10, line 223.

For a definitive answer, further external validations with large prospective studies are needed to confirm and compare the prognostic value of SUVLN / SUVTumor to that of SUVLN.

Comment #2

The inflamed tissue including tumor-draining LNs can also show high uptake of [18F]FDG. Did the uptake measurements for LNs verified by postoperative pathological examination to reduce the risk of false-positive results? Such data should be presented.

Reply #2

We agree with the reviewer’s comment that there is a possibility of false-positive findings on the clinically staged lymph nodes by FDG PET/CT. However, given the study patients were treated by definitive chemoradiotherapy (dCRT) rather than trimodality therapy (neoadjuvant CRT followed by surgery), we did not have postoperative pathological report to verify LNs with high FDG uptake as metastatic LNs. Nevertheless, to minimize the aforementioned problem, (refer to section 4.1. Patient selection, page 10, lines 246-249) the LNs were staged by contrast-enhanced CT and FDG PET/CT together in this study. Any discrepancies between the two imaging modalities results were resolved through the consensuses in multidisciplinary team meetings.

Comment #3
Table 2, why “Age” is shown in Bold font with underline?

Reply #3

Thank you for pointing out the error. We have corrected the error in the Table 2 in section 2.4. Correlations between parameters evaluated by FDG-PET and clinicopathological features, page 5, line 143

Round 2

Reviewer 2 Report

I can accept authors argument and believe the manuscript warrants publication.

Reviewer 3 Report

The authors adequately addressed this reviewer's concerns in this revised manuscript.